# Multi-Task Learning by Deep Collaboration and Application in Facial Landmark Detection

## Abstract

Convolutional neural networks (CNN) have become the most successful and popular approach in many vision-related domains. While CNNs are particularly well-suited for capturing a proper hierarchy of concepts from real-world images, they are limited to domains where data is abundant. Recent attempts have looked into mitigating this data scarcity problem by casting their original single-task problem into a new multi-task learning (MTL) problem. The main goal of this inductive transfer mechanism is to leverage domain-specific information from related tasks, in order to improve generalization on the main task. While recent results in the deep learning (DL) community have shown the promising potential of training task-specific CNNs in a soft parameter sharing framework, integrating the recent DL advances for improving knowledge sharing is still an open problem. In this paper, we propose the Deep Collaboration Network (DCNet), a novel approach for connecting task-specific CNNs in a MTL framework. We define connectivity in terms of two distinct non-linear transformation blocks. One aggregates task-specific features into global features, while the other merges back the global features with each task-specific network. Based on the observation that task relevance depends on depth, our transformation blocks use skip connections as suggested by residual network approaches, to more easily deactivate unrelated task-dependent features. To validate our approach, we employed facial landmark detection (FLD) datasets as they are readily amenable to MTL, given the number of tasks they include. Experimental results show that we can achieve up to 24.31% relative improvement in landmark failure rate over other state-of-the-art MTL approaches. We finally perform an ablation study showing that our approach effectively allows knowledge sharing, by leveraging domain-specific features at particular depths from tasks that we know are related.

## 1 Introduction

Over the past few years, convolutional neural networks (CNNs) have become the leading approach in many vision-related tasks (Krizhevsky et al., 2012). By creating a hierarchy of increasingly abstract concepts, they can transform complex high-dimensional input images into simple low-dimensional output features. Although CNNs are particularly well-suited for capturing a proper hierarchy of concepts from real-world images, successively training them requires large amount of data. Optimizing deep networks is tricky, not only because of problems like vanishing / exploding gradients (Hochreiter, 1998) or internal covariate shift (Ioffe & Szegedy, 2015), but also because they typically have many parameters to be learned (which can go up to 137 billions (Shazeer et al., 2017)). While previous works have looked at networks pre-trained on a large image-based dataset as a starting point for their gradient descent optimization, others have considered improving generalization by casting their original single-task problem into a new multi-task learning (MTL) problem (see Zhang & Yang (2017) for a review). As Caruana (1998) explained in his seminal work: "MTL improves generalization by leveraging the domain-specific information contained in the training signals of related tasks". Exploring new ways to efficiently gather more information from related tasks — the core contribution of our approach — can thus help a network to further improve upon its main task.

The use of MTL goes back several years, but has recently proven its value in several domains. As a consequence, it has become a dominant field of machine learning (Zhang & Zhou, 2014). Although many early and influential works contributed to this field (Evgeniou & Pontil, 2004), recent major advances in neural networks opened up opportunities for novel contributions in MTL. Works on grasping (Pinto & Gupta, 2017), pedestrian detection (Tian et al., 2015), natural language processing (Liu et al., 2015), face recognition (Yim et al., 2015; Yin & Liu, 2017) and object detection (Misra et al., 2016) have all shown that MTL has been finally adopted by the deep learning (DL) community as a way to mitigate the lack of data, and is thus growing in popularity.

MTL strategies can be divided into two major categories: *hard* and *soft* parameter sharing. Hard parameter sharing is the earliest and most common strategy for performing MTL, which dates back to the original work of Caruana (1998). Approaches in this category generally share the hidden layers between all tasks, while keeping separate outputs. Recent results in the DL community have shown that a central CNN with separate task-specific fully connected (FC) layers can successfully leverage domain-specific information (Ranjan et al., 2016; Zhang et al., 2014; Pinto & Gupta, 2017; Yin & Liu, 2017). Although hard parameter sharing reduces the risk of over-fitting (Baxter, 1997), shared layers are prone to be overwhelmed by features or contaminated by noise coming from particular noxious related tasks (Liu et al., 2017).

Soft parameter sharing has been proposed as an alternative to alleviate this drawback, and has been growing in popularity as a potential successor. Approaches in this category separate all hidden layers into task-specific models, while providing a knowledge sharing mechanism. Each model can then learn task-specific features without interfering with others, while still sharing their knowledge. Recent works using one network per task have looked at regularizing the distance between task-specific parameters with a $\ell_2$ norm (Duong et al., 2015) or a trace norm (Yang & Hospedales, 2016), training shared and private LSTM submodules (Liu et al., 2017), partitioning the hidden layers into subspaces (Ruder et al., 2017) and regularizing the FC layers with tensor normal priors (Long & Wang, 2015). In the domain of continual learning, progressive network (Rusu et al., 2016) has also shown promising results for cross-domain sequential transfer learning, by employing lateral connections to previously learned networks. Although all these soft parameter approaches have shown promising potential, improving the knowledge sharing mechanism is still an open problem.

In this paper, we thus present the deep collaboration network (DCNet), a novel approach for connecting task-specific networks in a *soft* parameter sharing MTL framework. We contribute with a novel knowledge sharing mechanism, dubbed the *collaborative block*, which implements connectivity in terms of two distinct non-linear transformations. One aggregates task-specific features into global features, and the other merges back the global features into each task-specific network. We demonstrate that our collaborative block can be dropped in any existing architectures as a whole, and can easily enable MTL for any approaches. We evaluated our method on the problem of facial landmark detection in a MTL framework and obtained better results in comparison to other approaches of the literature. We further assess the objectivity of our training framework by randomly varying the contribution of each related tasks, and finally give insights on how our collaborative block enables knowledge sharing with an ablation study on our DCNet.

The content of our paper is organized as follows. We first describe in Section 2 works on MTL closely related to our approach. We also describe Facial landmark detection, our targeted application. Architectural details of our proposed Multi-Task approach and its motivation are spelled out in Section 3. We then present in Section 4 a number of comparative results on this Facial landmark detection problem for two CNN architectures, AlexNet and ResNet18, that have been adapted with various MTL frameworks including ours. It also contains discussions on an ablation study showing at which depth feature maps from other tasks are borrowed to improve the main task. We conclude our paper in Section 5.

## 2 RELATED WORK

### 2.1 MULTI-TASK LEARNING

Our proposed deep collaboration network (DCNet) is related to other existing approaches. The first one is the cross-stitch (CS) (Misra et al., 2016) network, which connects task-specific networks through *linear* combinations of the spatial feature maps at specific layers. One drawback of CS is

that they are limited to capturing linear dependencies only, something we address in our proposed approach by employing *non-linearities* when sharing feature maps. Indeed, non-linear combinations are usually able to learn richer relationships, as demonstrated in deep networks. Another related approach is tasks-constrained deep convolutional network (TCDCN) for facial landmarks detection (Zhang et al., 2014). In it, the authors proposed an early-stopping criterion for removing auxiliary tasks before the network starts to over-fit to the detriment of the main task. One drawback of their approach is that their criterion has several hyper-parameters, which must all be selected manually. For instance, they define an hyper-parameter controlling the period length of the local window and a threshold that stops the task when the criterion exceeds it, all of which can be specified for each task independently. Unlike TCDCN, our approach has no hyper-parameters that depend on the tasks at hand, which greatly simplifies the training process. Our two transformation blocks consist of a series of batch normalization, ReLU, and convolutional layers shaped in a standard setting based on recent advances in residual network (see Sec. 3). This is particularly useful for computationally expensive deep networks, since integrating our proposed approach requires no additional hyper-parameter tuning experiments.

Our proposed approach is also related to HyperFace (Ranjan et al., 2016). In this work, the authors proposed to fuse the intermediate layers of AlexNet and exploit the hierarchical nature of the features. Their goal was to allow low-level features containing better localization properties to help tasks such as landmark localization and pose detection, and allow more class-specific high-level features to help tasks like face detection and gender recognition. Although HyperFace uses a single shared CNN instead of task-specific CNNs and is not entirely related to our approach, the idea of *feature fusion* is also central in our work. Instead of fusing the features at intermediate layers of a single CNN, our approach aggregates same-level features of multiple CNNs, at different depth independently. Also, one drawback of HyperFace is that the proposed feature fusion is specific to AlexNet, while our method is not specific to any network. In fact, our approach takes into account the vast diversity of existing network architectures, since it can be added to any architecture without modification.

## 2.2 FACIAL LANDMARK DETECTION

Facial landmark detection (FLD) is an essential component in many face-related tasks (Sun et al., 2013; Zhang et al., 2016; Jourabloo & Liu, 2016; Baltrušaitis et al., 2016). This problem can be described as follows: given the image of an individual's face, the goal is to predict the $(x, y)$-position on the image of specific landmarks associated with key features of the visage. Applications such as face recognition (Ding & Tao, 2015), face validation (Taigman et al., 2014), facial feature detection and tacking (Zhang & Zhang, 2014) rely on the ability to correctly find the location of these distinct facial landmarks in order to succeed. Localizing facial key points like the center of the eyes, the corners of the mouth, the tip of the nose and the earlobes is however a challenging problem when many lighting conditions, head poses, facial expressions and occlusions increase diversity of the face images. In addition to integrating this variability into the estimation process, a FLD model must also take into account a number of correlated factors. For instance, although both an angry person and a sad person have frowned eyebrows, an angry person will have pinched lips while a sad person will have sunken mouth corners (Fabian Benitez-Quiroz et al., 2016). A particularity of datasets geared towards FLD is that, on top of containing the position of these various facial landmarks, they also contain a number of other labels that can be seen as tasks on their own, such as gender recognition, smile recognition, glasses recognition or face orientation. For this reason, FLD datasets are particularly well-suited to evaluate MTL frameworks.

## 3 DEEP COLLABORATION

Given $T$ task-specific convolutional neural networks (CNNs), our goal is to share domain-specific information by connecting task-specific networks together using their respective feature maps. Unlike CS, our proposed approach will define this feature sharing in terms of two distinct non-linear transformations. Linear transformations like those used in CS can limit feature sharing ability, unlike ours that can learn complex transformations and properly connect each network.

For the sake of simplicity, we suppose that the networks have the same structure, which we refer to as the underlying network. Note that our approach also works with different architectures. Let us

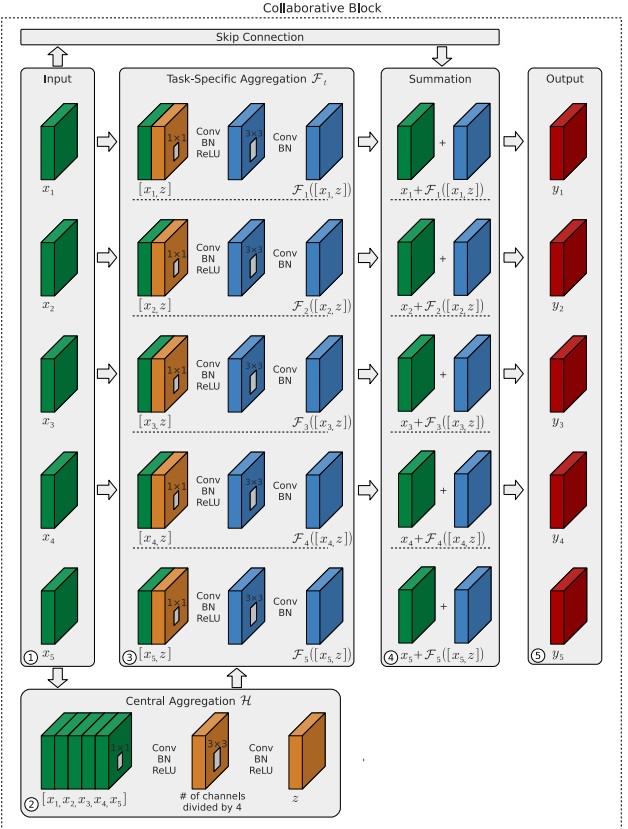

Figure 1: Example of our collaborative block applied on the feature maps of five task-specific networks. The input feature maps (shown in part 1) are first concatenated depth-wise and transformed into a global feature map (part 2). The global feature map is then concatenate with each input feature map individually and transformed into task-specific feature maps (part 3). Each resulting feature map is then added back to the input feature map using a skip connection (part 4), which gives the final outputs of the block (part 5).

further decompose the underlying network as a series of blocks, where each block can be as small as a single layer, as large as the whole network itself, or based on simple rules such as grouping all layers with matching spatial dimensions or grouping every $n$ subsequent layers. The arrangement of the layers into blocks does not change the nature of the network, but instead facilitates the understanding of applying our method. In particular, it makes explicit the depth at which we connect the feature maps via our framework.

Since our approach is independent on depth, we will drop the depth index on the feature maps to further simplify the equations. As such, we will define the feature map output of a block at a certain depth as $x_t$, where $t \in \{1 \dots T\}$, for each task $t$. Our approach takes as input all $x_t$ task-specific feature maps and processes them into new feature maps $y_t$ as follows:

$$z = \mathcal{H}([x_1, \dots, x_T]), \qquad y_t = x_t + \mathcal{F}_t([x_t, z]), \tag{1}$$

where $\mathcal{H}$ and $\mathcal{F}_t$ represent the central and the task-specific aggregations respectively, and $[\cdot]$ denotes depth-wise concatenation. We refer to (1) as our collaborative block. The goal of $\mathcal{H}$ is to combine all task-specific feature maps $x_t$ into a global feature map $z$ representing unified knowledge, while the goal of $\mathcal{F}$ is to merge back the global feature map $z$ with each input feature map $x_t$ individually. As shown in Fig. 1, $\mathcal{H}$ and $\mathcal{F}$ have the following structure:

$$\mathcal{H} = Conv_{(1\times1)} \circ BN \circ \delta \circ Conv_{(3\times3)} \circ BN \circ \delta, \tag{2}$$

$$\mathcal{F} = Conv_{(1\times1)} \circ BN \circ \delta \circ Conv_{(3\times3)} \circ BN, \tag{3}$$

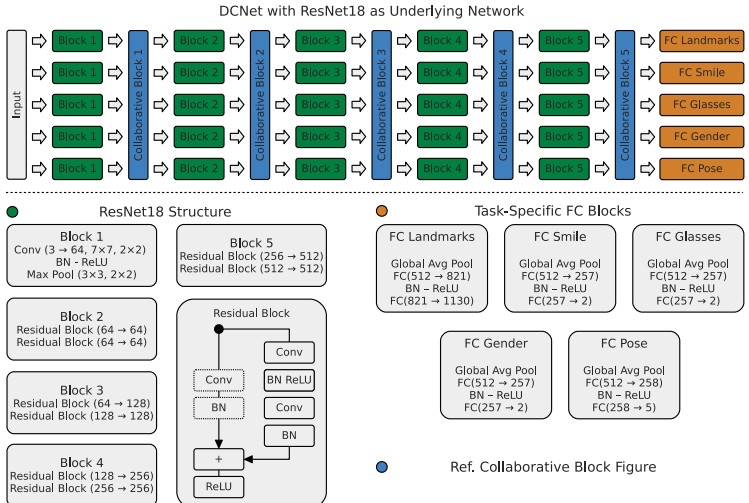

Figure 2: Deep Collaboration Network (DCNet) using ResNet18 as underlying network in a MTL setting on the MTFL dataset. The top part shows the block structure of ResNet18 interleaved with our proposed collaborative block. While the detailed composition of each ResNet block and the task-specific fully-connected blocks are shown at the bottom left and bottom right respectively, we refer to Fig. 1 for the description of our collaborative block.

where $BN$ stands for batch normalization, $\delta$ for the ReLU activation function and $Conv_{(h \times w)}$ for a standard convolutional layer with filters of size $(h \times w)$. The first $Conv_{(1 \times 1)}$ layer in $\mathcal{H}$ divides the number of feature maps by a factor of 4, while the first $Conv_{(1 \times 1)}$ layer in $\mathcal{F}$ divides it to match the size of $x_t$.

As seen by the presence of a skip connection in (1), the recent advances in residual network inspired the structure of our collaborative block. Based on the argument by He et al. (2016) that a network may more easily learn the proper underlying mapping by using an identity skip connection, we also argue that it may help our task-specific networks to more easily integrate the domain-information of each related task. One of the advantages of identity skip connections is that learning identity mappings can be done by simply pushing all parameters of the residual mapping towards zero. We integrated this observation in our collaborative block by allowing each task-specific network to easily remove the contribution of the global feature map $z$ with an identity skip connection, in order to account for cases where it does not help. As we see in (1), pushing all parameters of $\mathcal{F}_t$ towards zero would remove its contribution, and the output of the block would then simply revert to the original input $x_t$.

Our motivation for using an identity skip connection around the global feature map $z$ comes from the fact that the depth at which we insert our collaborative block influences the relevance of each task towards another. Considering that a network learns a hierarchy of increasingly abstract features, some task-specific networks may benefit more by sharing their low-level features than by sharing their high-level features. For instance, tasks such as landmark localization and pose detection may profit from their low-level features containing better localization properties, while tasks such as face detection and gender recognition may take advantage of their more class-specific high-level features. Our collaborative block can take into account task relevance by deactivating a different set of residual mappings $\mathcal{F}_t$ based on the level at which it appears in the network. An example of such specialization will be shown in our ablative study in Section 4.5.

Fig. 2 presents an example of our Deep Collaboration Network (DCNet) using ResNet18 as the underlying network. As we can see in the top part of the figure, this comes down to interleaving the underlying network block structure with our collaborative block. Each collaborative block receives as input the output of each task-specific block, processes them as detailed in Fig. 1, and sends the result back to each task-specific network. Adding our approach to any underlying network can be done by simply following the same pattern of interleaving the network block structure with our collaborative block.

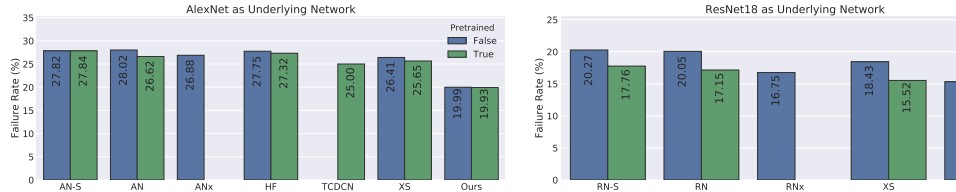

Figure 3: Landmark failure rates (%) on the MTFL task. The reported values are the average over the last five epochs, averaged over three tries. The left plot presents our results with AlexNet as the underlying network, while the right one with ResNet18. AN-S and RN-S stand for single-task training, AN and RN for multi-task training with a single central network, ANx and RNx for multi-task training with a single central network widen to match the number of parameters of our approach, HF for HyperFace, TCDCN for Zhang et al. (2014)'s approach and XS for Cross-Stitch. In each instance, the left column (blue) is for un-pretrained networks, while the right column (green) is for pre-trained networks. Our proposed approach obtains the lowest failure rates overall.

## 4 EXPERIMENTS

In this section, we detail our multi-task learning (MTL) training framework and present our experiments in facial landmark detection (FLD) tasks. We also analyze our approach by performing an ablation study and by experimenting with task importance.

### 4.1 MULTI-TASK LEARNING FRAMEWORK

As described previously, the problem of facial landmark detection is to predict the $(x, y)$-position on the image of specific landmarks associated with key features of the visage. While the number and type of landmarks are specific to each dataset, examples of standard landmarks to be predicted are the corners of the mouth, the tip of the nose and the center of the eyes. In addition to the facial landmarks, each dataset further defines a number of related tasks. These related tasks also vary from one dataset to another, and are typically gender recognition, smile recognition, glasses recognition or face orientation.

One particularity of our optimization framework is that we treat each task as a classification problem. While this is straightforward for gender, smile and glasses recognition as they are already classification tasks, it is a bit more tricky for face orientation and FLD. For face orientation, instead of predicting the roll, yaw and pitch real value as in a regression problem, we divide each component into 30 degrees wide bins and predict the label of the bin corresponding to the value. Similarly for FLD, rather than predicting the real $(x, y)$-position of each landmark, we divide the image into 1 pixel wide bins and predict the label of the bin corresponding to the value. Note that we still use the original real values when comparing our prediction with the ground truth, such that we incorporate our approximation errors in the final score. By doing this, do not artificially boost our performance.

We report our results using the landmark failure rate metric (Zhang et al., 2014), which is defined as follows: we first compute the mean distance between the predicted landmarks and the ground truth landmarks, then normalize it by the inter-ocular distance from the center of the eyes. A normalized mean distance greater than $10\%$ is reported as a failure.

### 4.2 MTFL FACIAL LANDMARK DETECTION

As a first experiment, we performed facial landmark detection on the Multi-Task Facial Landmark (MTFL) dataset (Zhang et al., 2014). The dataset contains 12,995 face images annotated with five facial landmarks and four related attributes of gender, smiling, wearing glasses and face profile (five profiles in total). The training set has 10,000 images, while the test set has 2,995 images. We perform four sets of experiments using an ImageNet pre-trained AlexNet, an ImageNet pre-trained ResNet18, an un-pretrained AlexNet and an un-pretrained ResNet18 as underlying networks. For AlexNet, we apply our collaborative block after each max pooling layer, while we do as shown in Fig. 2 for ResNet18.

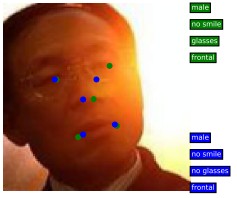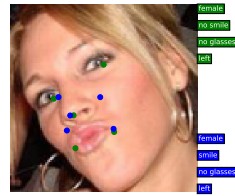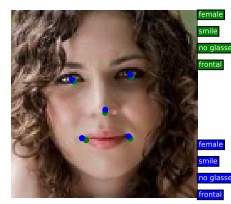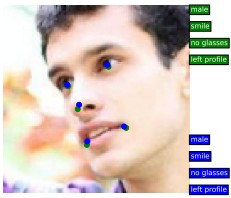

Figure 4: Example predictions of our DCNet with pre-trained ResNet18 as underlying network on the MTFL task. The first two contains failure cases, while last two contains successes. Elements in green correspond to ground truth, while those in blue correspond to our prediction. In addition to providing the facial landmarks (the small dots), we also include the labels of the related tasks: gender, smiling, wearing glasses and face profile. As shown in the first example, over-exposition can have a large impact on the prediction quality.

We compare our approach to several other approaches of the literature. We include single-task learning (AN-S when using AlexNet as underlying network, RN-S when using ResNet18), MTL with a single central network (AN and RN), MTL with a single central network that is widen to match the number of parameters of our approach (ANx and RNx), the HyperFace network (HF), the Tasks-Constrained Deep Convolutional Network (TCDCN) and the Cross-Stitch approach (CS). We train each network three times for 300 epochs and report landmark failure rates averaged over the last five epochs, further averaged over the three tries.

Fig. 3 presents our FLD results on the MTFL dataset. The left part of the figure corresponds to using AlexNet as underlying network, while the right one corresponds to ResNet18. For each case, the left column (blue) is for un-pretrained network, while the right column (green) is for ImageNet pre-trained network. Moreover, Fig. 4 shows example predictions from DCNet with pre-trained ResNet18 as underlying network. The first two examples were reported as failures, while the lsat two contains two successes. The ground truth elements are colored in green, while our predictions are colored in blue. In addition to showing facial landmarks as small dots, we also include the labels of the related tasks: gender, smiling, wearing glasses and face profile. The first example illustrates the influence of over-exposition on prediction.

One result we can observe from Fig. 3 is that our proposed approach obtained the lowest failure rates in each case. Indeed, our DCNet with un-pretrained and pre-trained AlexNet as underlying network obtained 19.99% and 19.93% failure rates respectively, while we obtained 15.32% and 14.34% with ResNet18. This is significantly lower than the other approaches to which we compare ourselves. For instance, with AlexNet, HF had 27.75% and 27.32%, XS had 26.41% and 25.65%, and TCDCN had 25.00%[1], while with ResNet18, XS had 18.43% and 15.52% respectively. We obtained the highest improvements when using AlexNet as the underlying network. With un-pretrained and pre-trained AlexNet, we obtained improvements of 6.42% and 5.07%, while we obtained 1.43% and 1.18% with ResNet18. Performing MTL with our approach can thus improve performance over using other approaches of the literature.

An interesting result from Fig. 3 is that although increasing the number of parameters of multi-task AlexNet (AN) and ResNet18 (RN) can significantly improve performance, connecting task-specific networks with our approach is more efficient. For instance, while AlexNet (ANx) and ResNet18 (RNx) with widened layers that match the number of parameters of our approach lowers the failure rates from 20.05% (RN) and 28.02% (AN) to 16.75% (RNx) and 26.88% (ANx) respectively, our approach with AlexNet and ResNet18 as underlying networks further reduces the failure rates to 15.32% and 19.99%. These results show that while simply increasing the number of parameters is an effortless avenue for improving performance, developing novel architectures enhancing network connectivity may open more rewarding research directions for further leveraging the domain-information of related tasks.

---

[1]Zhang *et. al* only provided results with pre-trained AlexNet (Zhang et al., 2014)

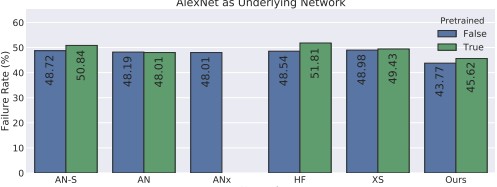 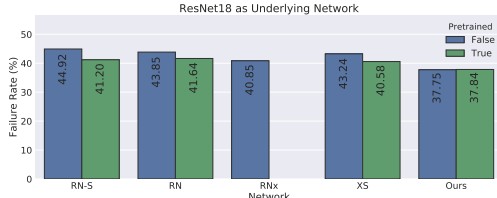

Figure 5: Landmark failure rates (%) on the AFW task. Each network is trained on the MTFL task and tested without fine-tuning on the AFW task. The left plot presents our results with AlexNet as the underlying network, while the right one with ResNet18. AN-S and RN-S stand for single-task learning, AN and RN for multi-task learning with a single central network, ANx and RNx for multi-task learning with a single central network widen to match the number of parameters of our approach, HF for HyperFace, TCDCN for Zhang et al. (2014)'s approach and XS for Cross-Stitch. In each instance, the left column (blue) is for un-pretrained networks, while the right column (green) is for pre-trained networks. Our proposed approach (last column) obtains the lowest failure rates overall.

## 4.3 AFW FACIAL LANDMARK DETECTION

As a second experiment, we performed domain adaptation on the Annotated Face in the Wild (AFW) dataset (Zhu & Ramanan, 2012). The dataset has 205 Flickr images, where each image can contain more than one face. Instead of using the images as provided, we process them using the available facial bounding boxes. We extract all faces with visible landmarks, which gives a total of 377 face images. We then take each network of Section 4.2 trained on the MTFL dataset and evaluate them without fine-tuning on these images. Fig. 5 presents the results of this experiment.

As it was the case in Section 4.2, our approach obtained the best results overall. Indeed, our DCNet with un-pretrained and pre-trained AlexNet as underlying network obtained 43.77% and 45.62% failure rates respectively, while it obtained 37.75% and 37.84% with ResNet18. This is significantly lower then the other approaches. For instance, with AlexNet, HF had 48.54% and 51.81%, and XS had 48.98% and 49.43%, while with ResNet18, XS had 43.24% and 40.58% respectively. Unlike what we observed in Section 4.2, our approach obtained the highest improvement with AlexNet when using an un-pretrained underlying network, while it obtained the highest improvement with ResNet18 when using a pre-trained underlying network. Indeed, DCNet with un-pretrained and pre-trained AlexNet obtained 4.24% and 2.39% improvements, while it obtained 3.10% and 2.74% with ResNet18 respectively.

An interesting result we can observe from Fig. 5 is that approaches with pre-trained AlexNet did not perform as well as those with pre-trained ResNet18, but rather increased the failure rates in comparison to the ones with un-pretrained AlexNet. For instance, single-task un-pretrained AlexNet obtained 48.72%, while single-task pre-trained AlexNet obtained a higher 50.84%. We also see a similar failure rate degradation for HF, XS and our approach. This is not the case when using ResNet18. Indeed, single-task un-pretrained ResNet18 obtained 44.92%, while single-task pre-trained ResNet18 obtained a lower 41.20%. Although the dataset is small and more experiments would help to better understand why this is happening, these results suggests that ResNet18 is more capable of adapting its pre-trained features for domain adaptation.

## 4.4 AFLW FACIAL LANDMARK DETECTION

As third experiment, we evaluate the influence of the number of training examples on MTL, using the Annotated Facial Landmarks in the Wild (AFLW) dataset (Koestinger et al., 2011). The dataset has 21,123 Flickr images, where each image can contain more than one face. Instead of using the images as provided, we process them using the available facial bounding boxes. We extract all faces with visible landmarks, which gives a total of 2,111 images. This dataset defines 21 facial landmarks and has 3 related tasks (gender, wearing glasses and face orientation). For face orientation, we divide the roll, yaw and pitch into 30 degrees wide bins (14 bins in total), and predict the label corresponding to each real value.

Table 1: Landmark failure rate results on the AFLW dataset using a pre-trained ResNet18 as underlying network. The presented values are averaged over the last five epochs, further averaged over three tries. The first column is the train / test ratio, and the subsequent ones are the networks: single-task ResNet18 (RN-S), multi-task ResNet18 (RN) and Cross-Stitch network (CS). In all cases except the first one, our approach obtains the best performance.

| Train / Test Ratio | Networks | | | |
|---|---|---|---|---|
| | RN-S | RN | XS | Ours |
| 0.1 / 0.9 | **57.39** | 58.00 | 73.06 | 60.20 |
| 0.3 / 0.7 | 31.84 | 32.00 | 36.24 | **29.84** |
| 0.5 / 0.5 | 23.41 | 23.31 | 26.02 | **21.84** |
| 0.7 / 0.3 | 21.47 | 21.92 | 22.37 | **18.42** |
| 0.9 / 0.1 | 13.03 | 12.80 | 13.51 | **11.32** |

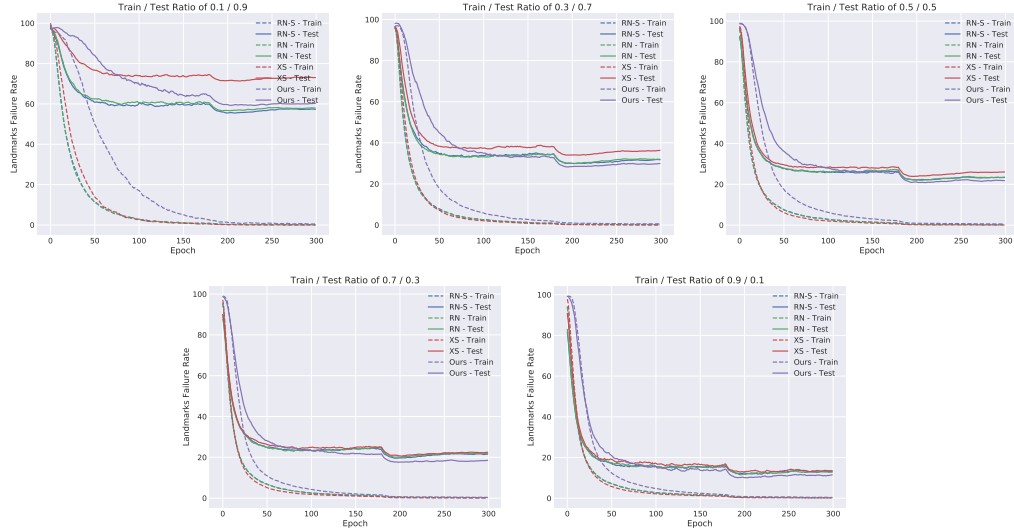

Figure 6: Landmark failure rate progression (in %) on the AFLW dataset with varying train / test ratio using a pre-trained ResNet18 as base network. Each curve is the average over three tries. Even though our approach has the slowest convergence rate, it outperforms the others in four of the five cases.

Our experiment works as follows. With a pre-trained ResNet18 as underlying network, we compare our approach to single-task ResNet18 (RN-S), multi-task ResNet18 (RN) and Cross-Stitch network (XS) by training on a varying number of images. We use five different train / test ratios, starting with 0.1 / 0.9 up to 0.9 / 0.1 by 0.2 increment. In other words, we train each approaches on the first 10% of the available images and test on the other 90%, then repeat for all the other train / test ratios. We use the same training framework as in section 4.2. We train each network three times for 300 epochs, and report the landmark failure rate averaged over the last five epochs, further averaged over the three tries.

As we can see in Table 1, our approach obtained the best performance in all cases except the first one. Indeed, we observe between 1.46% and 3.05% improvements with train / test ratios from 0.3 / 0.7 to 0.9 / 0.1, while we obtain a negative relative change of 4.90% with train / test ratio of 0.1 / 0.9. In fact, since all multi-task approaches obtained higher failure rates than the single-task approach, this suggests that the networks are over-fitting the small training set. Nonetheless, these results show that we can obtain better performance using our approach.

Figure 6 presents the landmark failure rate progression on all train / test ratios for each network. One interesting result we can see from these figures is that our approach has the slowest convergence rate during the first stage of the training process. For instance, with a train / test ratio of 0.9 / 0.1, our

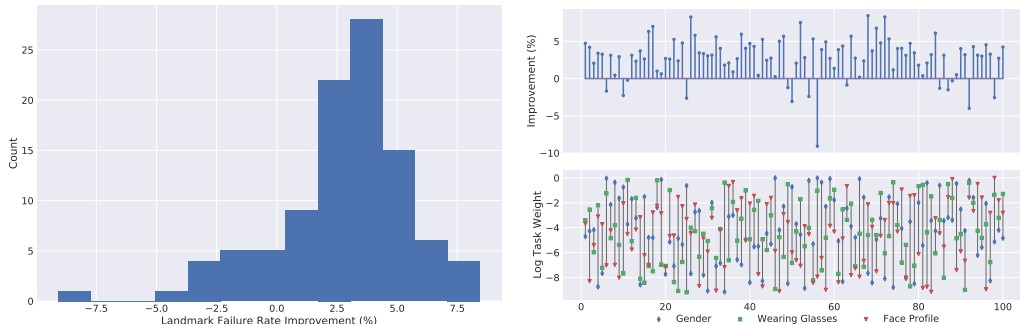

Figure 7: Landmark failure rate improvement (in %) of our approach compared to XS when sampling random task weights. We used a pre-trained ResNet18 as underlying network. The histogram at the left and the plot at the top right represents performance improvement achieved by our proposed approach (positive value means lower failure rates), while the plot at the bottom right corresponds to the log of the task weights. Our approach outperformed XS in 86 out of the 100 tries, thus empirically demonstrating that our learning framework was not unfavorable towards XS and that our approach is less sensitive to the task weights $\lambda$.

approach converges at about epoch 100, while the others start converging at about epoch 50. In fact, while the other methods have similar convergence rate, the epoch at which our approach converges increases as the number of training images decrease. Indeed, our approach converges at about epoch 130 with a train / test ratio of 0.5 / 0.5, while it converges at about epoch 160 with a train / test ratio of 0.1 / 0.9. Even though the convergence rate is slower, our approach still converges to a similar train failure rate. This gives a smaller train-to-test gap, which indicates that our approach has better generalization abilities.

One particularity that we observe in Table 1 is that the XS network has relatively high failure rates. In the previous experiments of sections 4.2 and 4.3, XS had either similar or better performance than the other approaches (except ours). This could be due to our current multi-task learning framework that is unfavorable towards XS, which may prevent it from leveraging the domain-information of the related tasks. In order to investigate whether this is the case, we perform the following additional experiment. Using a pre-trained ResNet18 as underlying network, we compare our approach to XS by training each network 100 times using task weights randomly sampled from a log-uniform distribution. Specifically, we first sample from a uniform distribution $\gamma \sim \mathcal{U}(\log(1e-4), \log(1))$, then use $\lambda = \exp(\gamma)$ as the weight. We trained both XS and our approach for 300 epochs with the same task weights using a train / test ratio of 0.5 / 0.5.

Figure 7 presents the results of this experiment. The plot at the top right of the figure represents the landmark failure rate improvement (in %) of our approach compared to XS, while the plot at the bottom right corresponds to the log of the task weights for each try. In 86 out of the 100 tries, our approach had a positive failure rate improvement, that is, obtained lower failure rates than XS. Moreover, as we can see in the histogram at the left of Fig. 7, in addition being normally distributed around a mean of 2.78%, our approach has a median failure rate improvement of 3.14% and a maximum improvement of 8.45%. These results show that even though we sampled at random different weights for the related tasks, our approach outperforms XS in the majority of the cases. Our learning framework is therefore not unfavorable toward XS.

## 4.5 ABLATION STUDY

As a final experiment, we perform an ablation study on our approach using the MTFL dataset with an un-pretrained ResNet18 as base network. The goal of this experiment is to test whether the observed performance improvements is due to the increased ability of each task-specific network to share their respective features, rather than only due to the intrinsic added representative ability of using additional non-linear layers. Our experiment works as follows: we evaluate the impact of removing the contribution of each task-specific network by masking their respective feature maps at the input of the central aggregation transformation. By referring to Fig. 1, this corresponds to

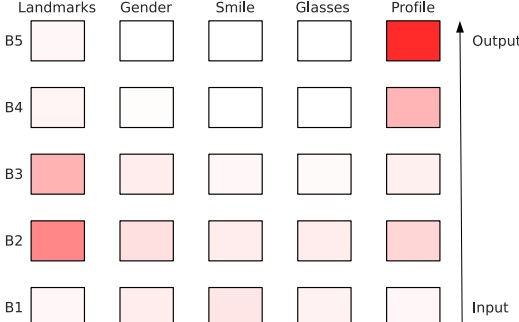

Figure 8: Results of our ablation study on the MTFL dataset with an un-pretrained ResNet18 as underlying network. We remove each task-specific features from each respective central aggregation layer and evaluate the effect on landmark failure rate. The columns represent the task-specific network, while the rows correspond to the network block structure. Blocks with a high saturated color were found to have a large impact on performance. For instance, this ablative study shows that the influence of high-level face profile features is large within our proposed architecture, which corroborates with the well-known fact that facial landmark locations are highly correlated to profile orientation. This thus constitutes an empirical evidence of domain-specific information sharing via our approach.

zeroing out the designated feature maps during concatenation in part 2 of the collaborative block (at the bottom of the figure). Note that the collaborative ResNet18 is trained on the MTFL dataset using the same framework as explained in Sec. 4.1, and the ablation study is performed on the test set.

Figure 8 presents the results of our ablation study. The columns represent each task-specific network, while the rows correspond to the network block structure. The blocks are ordered from bottom (input) to top (prediction), while the task-specific networks are ordered from left (main task) to right (related tasks). The color saturation indicates the influence of removing the task-specific feature maps from the corresponding central aggregation. A high saturation reflects high influence, while a low saturation reflects low influence.

A first interesting result that we can see from Fig. 8 is that removing features from the facial landmark detection network significantly increases landmark failure rate. For instance, we observe a negative (worse) relative change of 29.72% and 47.00% in failure rate by removing features from B3 and B2 respectively. This is interesting, as it shows that the main-task network contributes to and feed from the global features computed by the central aggregation transformation. Note that due to using a skip connection between the input and the global features, the network can remove the contribution of the global features by simply zeroing out its task-specific aggregation weights. These results show that the opposite is instead happening, where the task-specific features from the facial landmark network influence the quality of the computed global features, which in turn influence the quality of the subsequent task-specific features.

Another interesting result is that B5 from the face profile-related task has the highest influence on failure rate. Indeed, we observe a negative relative change of 83.87% by removing the corresponding features maps from the central aggregation. Knowing that face orientation has a direct impact on the location of the facial landmarks, it makes sense that features from the head related task would be useful for improving landmark predictions. What is interesting in this case is that B5 has the highest-level features of all blocks, due to being at the top of the hierarchy of increasingly abstract features. Since we expect the highest-level features of the head network to resemble a standard rotation matrix, it is evident that the landmark network would use these rich features to better rotate the predicted facial landmarks. This is what we observe in Fig. 8. These results constitute an empirical evidence that our approach not only allows leveraging domain-specific information from related tasks, but also improves knowledge sharing with better network connectivity.

## 5    CONCLUSION AND FUTURE WORK

In this paper, we proposed the deep collaboration network (DCNet), a novel approach for connecting task-specific networks in a multi-task learning setting. It implements feature connectivity and sharing through two distinct non-linear transformations inside a collaborative block, which also incorporates skip connection and residual mapping that are known for their good training behavior. The first transformation aggregates the task-specific feature maps into a global feature map representing unified knowledge, and the second one merges it back into each task-specific network. One key characteristic of our collaborative blocks is that they can be dropped in virtually any existing architectures, making them universal adapters to endow deep networks with multi-task learning capabilities.

Our results on the MTFL, AFW and AFLW datasets showed that our DCNet outperformed several state-of-the-art approaches, including cross-stitch networks. Our additional ablation study, using ResNet18 as underlying network, confirmed our intuition that the task-specific networks exploited the added flexibility provided by our approach. Additionally, these task-specific networks successfully incorporated features having varying levels of abstraction. Evaluating our proposed approach on other MTL problems could be an interesting avenue for future works. For instance, the recurrent networks used to solve natural language processing problems could benefit from incorporating our novel method leveraging domain-information of related tasks.

### ACKNOWLEDGEMENTS

We gratefully acknowledge the support of NVIDIA Corporation for providing a Tesla Titan X for our experiments through their Hardware Grant Program.

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
