# OpenReview forum: "Multi-Task Learning by Deep Collaboration and Application in Facial Landmark Detection"
_ICLR.cc/2018/Conference — Reject_

### Official Review · AnonReviewer2 · 2017-11-27
**This paper proposed a new block to combine domain-specific information from related tasks, in order to improve generalization of the target tasks. Although the relative improvement seems high (24.31%), its novelty is a little limited, and the target task in this submission(5 landmarks detection) is too simple to prove the effectiveness.**

**Rating:** 5
**Confidence:** 5

**Review:**

Pros:
1. This paper proposed a new block which can aggregate features from different tasks. By doing this, it can take advantage of common information between related tasks and improve the generalization of target tasks.

2. The achievement in this paper seems good, which is 24.31%.

Cons:
1. The novelty of this submission seems a little limited.

2. The target task utilized in this paper is too simple, which only detects 5 facial landmarks. It is hard to say this proposed work can still work when facing more challenging tasks, for example, 60+ facial landmarks prediction.

3. " Also, one drawback of HyperFace is that the proposed feature fusion is specific to AlexNet," In the original submission, HyperFace is based on AlexNet, but does this mean it can only work on AlexNet?

---

> ### Author Response · Authors · 2017-12-21
> **Answer to AnonReviewer2**
>
> 1. The novelty of this submission seems a little limited.
>
> Several advances in deep learning from the past 5 years have shown that simple approaches sometimes yield large improvements. One of the most striking example is the use of identity skip connection in Residual Network. The novelty of simply adding the feature map at a lower layer to the feature map at a higher layer could be viewed as limited. Other contributions that could be seen as limited also include batch normalisation, the  squeeze and excitation block from the winners of ImageNet 2017 competition (feature map calibration by global average pooling and scaling), DenseNet (change the sum in ResNet by a concatenation) and even the ReLU. The novelty of all these approaches may seem limited, but this is because they are fairly simple to understand.
>
> One crucial advantage of simple contributions is the possibility of straightforward integration into existing projects. As an example, AnonReviewer3 asked us to include MTCNN in our experiments. We found the following Tensorflow github project https://github.com/AITTSMD/MTCNN-Tensorflow that implements and trains MTCNN. It took use little time to implement our collaborative block in Tensorflow, and integrate it to their network. We have included our code at the end of our comment. This is a key advantage of our contribution, that it can be incorporated into existing projects without difficulties.
>
> # Tensorflow python implementation of our collaborative block
>
> def collaborative_block(inputs, nf, training, scope):
>     def conv2d(x, nf, fs):
>         y = slim.conv2d(x, nf, [fs, fs], 1, 'SAME', activation_fn=None,
>                         weights_initializer=slim.xavier_initializer(),
>                         biases_initializer=None,
>                         weights_regularizer=slim.l2_regularizer(0.0005))
>         return y
>
>     def bn(x, training):
>         y = slim.batch_norm(x, 0.999, True, True, 1e-5, is_training=training)
>          return y
>
>     def aggregation(x, n_out, training):
>         with tf.variable_scope('aggregation'):
>             z = x
>             z = conv2d(z, n_out, 1)
>             z = bn(z, training)
>             z = tf.nn.relu(z)
>             z = conv2d(z, n_out, 3)
>             z = bn(z, training)
>         return z
>
>     def central_aggregation(inputs, n_out, training):
>         with tf.variable_scope('central'):
>             z = tf.concat(inputs, axis=-1)
>             z = aggregation(z, n_out, training)
>             z = tf.nn.relu(z)
>         return z
>
>     def local_aggregation(x, z, n_out, pos, training):
>         with tf.variable_scope('local_{}'.format(pos)):
>             y = tf.concat([x, z], axis=-1)
>             y = x + aggregation(y, n_out, training)
>         return y
>
>     with tf.variable_scope(scope):
>         n_inputs = len(inputs)
>         z = central_aggregation(inputs, nf * n_inputs // 4, training)
>         outputs = [local_aggregation(x, z, nf, i, training)
>                    for i, x in enumerate(inputs)]
>     return outputs
>
>
> 2. The target task utilized in this paper is too simple, which only detects 5 facial landmarks. It is hard to say this proposed work can still work when facing more challenging tasks, for example, 60+ facial landmarks prediction.
>
> It is true that we only predict 5 facial landmarks in our experiments on the MTFL and AFW datasets. However, we predict 21 facial landmarks in our experiment on the AFLW dataset. This is a substantially harder task. As we write in table 1 in section 4.4, the results show that our approach outperforms both the standard multi-task setting (hard-parameter sharing) and the cross-stitch approach (soft-parameter sharing).
>
>
> 3. "Also, one drawback of HyperFace is that the proposed feature fusion is specific to AlexNet," In the original submission, HyperFace is based on AlexNet, but does this mean it can only work on AlexNet?
>
> In the version of their paper what we read on arXiv, i.e. version 2, they wrote that they propose “a novel CNN architecture” as one of their contributions. They did not elaborate on how to adapt their approach on other network architectures. At that time, it was not clear for us how to make it work on other network. However, the authors recently (December 6, 2017) updated their arXiv paper to version 3, with a network architecture based on residual connections (see https://arxiv.org/abs/1603.01249). They call it the HyperFace-ResNet, in contrast to their original HyperFace network using AlexNet. Since now they show how to adapt their approach to another network, we agree that it is no longer a drawback that it is only specific to AlexNet. We will therefore remove our sentence.

---

### Official Review · AnonReviewer3 · 2017-11-28
**Interesting method; better to do comparison with previous methods**

**Rating:** 6
**Confidence:** 4

**Review:**



This paper proposes a multi-pathway neural network for facial landmark detection with multitask learning. In particular, each pathway corresponds to one task, and the intermediate features are fused at multiple layers. The fused features are added to the task-specific pathway using a residual connection (the input of the residual connection are the concatenation of the task-specific features and the fuse features). The residual connection allows each pathway to selectively use the information from other pathways and focus on its own task.

This paper is well written. The proposed neural network architectures are reasonable.

The residual connection can help each pathway to focus on its own task (suggested by Figure 8). This phenomenon is not guaranteed by the training objective but happens automatically due to the architecture, which is interesting.

The proposed model outperforms several baseline models. On MTFL, when using the AlexNet, the improvement is significant; when using the ResNet18, the improvement is encouraging but not so significant. On AFLW (trained on MTFL), the improvements are significant in both cases.

What is missing is the comparison with other methods (besides the baseline). For examples, it will be helpful to compare with existing non-multitask learning methods, like TCDCN (Zhang et al., 2014) (it seems to achieve 25% failure rate on AFLW, which is lower than the numbers in Figure 5), and  multi-task learning method, like MTCNN (Zhang et al., 2016). It is important to show that proposed multitask learning method is useful in practice.
In addition, many papers take the average error as the performance metric. Providing results in the average error can make the experiments more comprehensive.

The proposed architecture is a bit huge. It scales linearly with the number of tasks, which is not quite preferable. It is also not straightforward to add new tasks to finetune a trained model.

In Figure 5 (left), it is a bit weird that the pretrained model underperforms the nonpretrained one.

I am likely to change the rating based on the comparison with other methods.

---

> ### Author Response · Authors · 2017-12-21
> **Answer to AnonReviewer3**
>
> 1. What is missing is the comparison with other methods...
>
> We would like to first clarify some elements that seem to be confusing. The authors of the TCDCN approach, Zhang et al., 2014, first introduced the MTFL dataset in their paper ( http://mmlab.ie.cuhk.edu.hk/projects/TCDCN.html). This dataset was created by merging two datasets together, one for training and one for testing. The training is Sun’s et al. dataset, which itself is constituted of two datasets: LFW dataset and their proposed Net dataset. As for the test set, Zhang et al., 2014 randomly selected 3000 images from the AFLW dataset.
>
> The performance of 25% by TCDCN therefore correspond to the accuracy on the test set of MTFL. As can be seen in Figure 3, we already compared ourselves to TCDCN.
>
> The results in Figure 5 are for a different test set. We still train on the training dataset of MTFL (LFW + Net), but this time we test on the AFW dataset from Zhu et al. 2012.
>
> We would have liked to compare ourselves to TCDCN on AlexNet (initialized at random) and ResNet, but the authors did not yet provide the training code. There is an open issue in their Github project: https://github.com/zhzhanp/TCDCN-face-alignment/issues/7
>
> Regarding MTCNN, we are currently running experiments using the MTCNN-Tensorflow github project. We implemented our collaborative block and incorporated it to the network. The results should be available in the following 2 weeks or so, but already look promising.
>
> 2. In addition, many papers take the average error as the performance metric.
>
> We did not include the average error since we thought the metric error was sufficient. However, we agree that it would be more comprehensive to show them. We will put them in a following comment.
>
> 3. The proposed architecture is a bit huge...
>
> Our main contribution is the collaborative block, which connects task-specific networks in a soft parameter sharing MTL setting. The linear increase with the number of tasks is a limitation of this setting, which is well-known in the MTL community. The effective increase of our collaborative block is in itself limited two conv layers for the central aggregation, and two conv layers for each task-specific aggregation.
>
> When using ResNet18 as underlying network, we used 5 collaborative blocks. With 4 tasks in total, the size of each block (in order of depth) was 234,752, 234,752, 936,448, 3,740,672 and 14,952,448. The single-task ResNet18 has 11,176,512 parameters, so the five tasks soft-parameter network has 55,882,560 parameters. For ResNet18, that increase may be large, but note that it does not scale with depth. Using a ResNet101 with 44,549,160 parameters, the five tasks soft-parameter network would have 222,745,800. The relative parameter increase of our approach would be lower.
>
> 4. It is also not straightforward to add new tasks to finetune a trained model.
>
> In our soft-parameter sharing multi-task setting, finetuning on a new task can be done by simply connecting a new task-specific network to the other networks. In the case where we do not have access (during finetuning) to the original tasks on which the network was trained on, it is always possible to freeze the weights of the pre-trained task-specific networks. This has the advantage of avoiding catastrophic forgetting, where the features learned from the previous tasks are removed during finetuning. This is in contrast with hard-parameter sharing, where only the fully connected layers are separated. In that case, the network must be finetuned using all previous tasks, otherwise the shared intermediate layers can experiment catastrophic forgetting.
>
> 5. In Figure 5 (left), it is a bit weird that the pretrained model underperforms the nonpretrained one.
>
> For this experiment, the networks were pretrained on ImageNet, fine-tuned on MTFL, then tested on AFW. In other words, the networks were pretrained on a first domain, finetuned on a second domain then tested on a third domain. We believe that, in order to develop good domain adaptation abilities during finetuning, it was harder for the networks to adjust its features learned on ImageNet than learned them from a random initialisation start.

---

> > ### Author Response · Authors · 2017-12-21
> > **Average error performance**
> >
> > These are the values that we obtained (in %):
> >
> > 1. MTFL
> >
> > 1.1 Underlying Network: AlexNet
> >
> > 1.1.1 Not pre-trained
> > AN-S: 9.474, AN: 9.435, ANx: 9.356, HF: 9.548, TCDCN: unknown, XS: 9.379, Ours: 8.423
> >
> > 1.1.2 Pre-trained
> > AN-S: 9.473, AN: 9.395, HF: 9.426, TCDCN: unknown, XS: 9.377, Ours: 8.500
> >
> > 1.2 Underlying Network: ResNet
> >
> > 1.2.1 Not pre-trained
> > RN-S: 8.692, RN: 8.571, RNx: 8.236, XS: 8.456, Ours: 8.007
> >
> > 1.2.2 Pre-trained
> > RN-S: 8.262, RN: 8.170, XS: 7.953, Ours: 7.845
> >
> > 2. AFW
> >
> > 2.1 Underlying Network: AlexNet
> >
> > 2.1.1 Not pre-trained
> > AN-S: 16.04, AN: 16.06, ANx: 16.68, HF: 16.34, XS: 18.24, Ours: 15.14
> >
> > 2.1.2 Pre-trained
> > AN-S: 17.24, AN: 17.15, HF: 16.34, XS: 17.78, Ours: 16.95
> >
> > 2.2 Underlying Network: ResNet
> >
> > 2.2.1 Not pre-trained
> > RN-S: 16.15, RN: 15.71, RNx: 14.81, XS: 15.78, Ours: 14.14
> >
> > 2.2.2 Pre-trained
> > RN-S: 14.73, RN: 14.97, XS: 15.70, Ours: 14.27

---

> > > ### Comment · AnonReviewer3 · 2017-12-22
> > > **Thanks for the response**
> > >
> > > Thank you for the detailed clarification.
> > >
> > > For the test settings, thank you for the clarification. It is clear now.
> > > I agree that it is hard to fully replicate previous methods and retrain the models with an up-to-date neural network. However, without that results, the experiments seem a bit not solid enough.
> > > The explanations of the other limitations are reasonable, but they do not resolve the actual limitation. I would like to just take these limitations.
> > >
> > > Some of my concerns are addressed, and there is a chance for the authors to provide an updated MTCNN results. I would like to slightly increase the rating.

---

> > > > ### Author Response · Authors · 2018-01-04
> > > > **MTCNN results**
> > > >
> > > > Here are some preliminary results of our approach on MTCNN. Note that due to the time allotted,  we did not fully explored all variations of our training process. We are currently performing more experiments, especially on the structure of the underlying networks PNet, RNet and ONet, and on the hard negative data generation process (we used the provided hyper-parameters from the authors of the MTCNN-Tensorflow github project, which may have been fine-tuned for the original MTCNN).
> > > >
> > > > We used as training datasets Sun’s et al. dataset (LFW+Net) for landmarks and the Wider face dataset for face recognition. We test on the Celeba test set, which contains 19962 images. Here are the results that we obtained:
> > > >
> > > > MTCNN original (ran ourselves)
> > > >
> > > >     Number of face detection failures: 44
> > > >     Mean dists: 8.1124
> > > >     Median dists: 3.9281
> > > >     Mean landmark failure rate: 0.2201
> > > >
> > > > Ours
> > > >
> > > >     Number of face detection failures: 16
> > > >     Mean dists: 8.5217
> > > >     Median dists: 2.8076
> > > >     Mean landmark failure rate: 0.1258
> > > >
> > > > With our approach, MTCNN has fewer face detection failures (16 vs 44) and a lower landmark failure rate (0.1258 vs 0.2201). The mean distance is however larger (8.5217 vs 8.1124), but the median is lower (2.8076 vs  3.9281). We are currently running experiments on the MTFL dataset to see if we obtain similar improvements.

---

### Official Review · AnonReviewer1 · 2017-11-29
**The authors propose a collaborative block that can be inserted in any deep network for multi-task learning and evaluate the method on multiple tasks related to Faces**

**Rating:** 6
**Confidence:** 4

**Review:**

The collaborative block that authors propose is a generalized module that can be inserted in deep architectures for better multi-task learning. The problem is relevant as we are pushing deep networks to learn representation for multiple tasks. The proposed method while simple is novel. The few places where the paper needs improvement are:

1. The authors should test their collaborative block on multiple tasks where the tasks are less related. Ex: Scene and object classification. The current datasets where the model is evaluated is limited to Faces which is a constrained setting. It would be great if Authors provide more experiments beyond Faces to test the universality of the proposed approach.
2. The Face datasets are rather small. I wonder if the accuracy improvements hold on larger datasets and if authors can comment on any large scale experiments they have done using the proposed architecture.

In it's current form I would say the experiment section and large scale experiments are two places where the paper falls short.

---

> ### Author Response · Authors · 2017-12-21
> **Answer to AnonReviewer1**
>
> 1.  We decided to perform our multi-task experiments on facial landmark detection because several previous approaches have shown that training on face orientation regression, along with gender, smile and glasses classification, can help better detect facial landmarks. We wanted to first demonstrate that our approach could leverage domain-specific information from tasks that we know were related. In particular, this allowed our ablation study in section 4.5 to provide empirical and easily interpretable evidence that our approach could indeed take advantage of high level face profile features to boost facial landmark detection.
>
> However, we agree that including experiments in other domains would further improve diversity. In that sense, we started working immediately after submission on tasks unrelated to faces, to precisely test the universality of our approach. So far, all conducted experiments were positive. For instance, we have an ongoing project on tree species identification from images of bark. This yet unpublished dataset contains 750,000+ unique crops from close-up pictures of bark for 22 different tree species, along with their trunk diameter (DBH). This is a multi-task setting where the tasks are less related. Indeed, different types of trees can have the same DBH, and inversely, trees from the same category can have different DBH (reflecting for instance their age). Using our approach, we could improve the classification accuracy from 93,09% to 94,46%. We can add this result in a new section to provide additional evidence that our approach can also work in a multi-task setting where tasks are less related.
>
> We also looked at using our collaborative block on a standard object recognition problem. Instead of connecting task-specific networks to perform multi-task learning, we create a network by repeating our collaborative block to perform single-task learning. The network processes the input image using multiple connected branches, and outputs a feature vector at the end of the convolutional layers (before the fully connected layer) that is the concatenation of the features computed by each branch. In this setting, our approach goes in line with current works that try to alleviate large processing time by trading depth for width, i.e. by using fewer layers with more weights. Our current preliminary results have shown that increasing width by having more collaborative branches is an effective way to achieve similar error rates, while using fewer weights. On the CIFAR-10 dataset, with standard data augmentation (horizontal flip and ±4 pixels translation), we obtained 3.96% classification error using only 6,988,986 parameters. In comparison to recent approaches that explored trading depth for width, our approach has the lowest number of weights (relatively to obtaining around 4% error rate), as seen below:
>
> Wide ResNet
> 4.00% with 36,479,194 parameters (https://github.com/szagoruyko/wide-residual-networks)
>
> ResNeXt
> 4.00% with around 9.2M parameters (estimated from the curve in Fig. 7 of their arxiv paper https://arxiv.org/pdf/1611.05431.pdf)
>
> AOGNet-BN
> 3.99 with 8.0M parameters (https://arxiv.org/pdf/1711.05847.pdf)
>
> Ours:
> 3.96 % with 6,988,986 parameters
>
> We are currently performing more experiments, but we could include this result on CIFAR-10.
>
>
> 2.  We started with smaller datasets because we wanted to test several networks. For instance, the results in Figure 5 took us around one month to obtain with our single GPU architecture. This is because we used two underlying networks, which can either be pre-trained or not, and trained them in either a single-task setting or one of the different multi-task settings. We wanted to compare our approach to many other networks before going on larger datasets.
>
> However, we agree that it would be best to have larger datasets. In that sense, our current work on multi-task tree classification mentioned above could be considered a more large scale experiment, as it contains 750,000+ unique crops of 224x224 pixels. Moreover, we are currently implementing our approach on the MTCNN facial landmark detection approach. The network is trained to perform both face detection and landmarks detection. For face detection, we are using the WIDER dataset (http://mmlab.ie.cuhk.edu.hk/projects/WIDERFace/) containing 393,703 face images, and for facial landmark detection, we are using the celebA dataset (http://mmlab.ie.cuhk.edu.hk/projects/CelebA.html) containing 202,599 face images. The results of this experiment should be available in the next 2 weeks or so, but are already looking promising. See my answer to AnonReviewer3 for more details.

---

### Decision · Program_Chairs · 2018-01-29
**ICLR 2018 Conference Acceptance Decision**

**Decision:**

Reject

**Comment:**

The experimental work in this paper leaves it just short of being suitable for acceptance.
The work needs more comparisons with prior work and other approaches.
The numerical ratings of the work by reviewers are just too low.